# Discovery of Hydrazineyl Amide Derivative of Pseudolaric Acid B for Reprogramming Tumor-Associated Macrophages Against Tumor Growth

**DOI:** 10.3390/molecules30102088

**Published:** 2025-05-08

**Authors:** Xia Peng, Siqi Yu, Lin Xu, Qinghua Wang, Lin Yang, Yi Su, Zhirou Xiong, Mengjie Shao, Meiyu Geng, Ao Zhang, Lei Zhang, Jing Ai, Chunyong Ding

**Affiliations:** 1School of Chinese Materia Medica, Nanjing University of Chinese Medicine, Nanjing 210023, China; xpeng@simm.ac.cn (X.P.); ysu@simm.ac.cn (Y.S.); mygeng@simm.ac.cn (M.G.); 2State Key Laboratory of Drug Research, Shanghai Institute of Materia Medica, Chinese Academy of Sciences, Shanghai 201203, China; xiongzhirou@simm.ac.cn; 3School of Pharmaceutical Sciences, University of Chinese Academy of Sciences, Beijing 100049, China; 4Shanghai Frontiers Science Center of Drug Target Identification and Delivery, National Key Laboratory of Innovative Immunotherapy, School of Pharmaceutical Sciences, Shanghai Jiao Tong University, Shanghai 200240, China; y13419229737@163.com (S.Y.); 15772173438@163.com (L.X.); wqh233333@126.com (Q.W.); m13727007061@163.com (L.Y.); ao6919zhang@sjtu.edu.cn (A.Z.); 5School of Pharmaceutical Sciences, Zunyi Medical University, Zunyi 563000, China; 6Shandong Laboratory of Yantai Drug Discovery, Bohai Rim Advanced Research Institute for Drug Discovery, Yantai 264117, China; 7Key Laboratory for Modernization of Qiandongnan Miao & Dong Medicine, Qiandongnan Traditional Medicine Research & Development Center, School of Life and Health Science, Kaili University, Kaili 556011, China

**Keywords:** pseudolaric acid B, amide derivatives, tumor-associated macrophages, anti-tumor, repolarization

## Abstract

Tumor-associated macrophages (TAMs) are pivotal for tumor development and progression. Reprogramming the M2-like pro-tumoral behavior of TAMs towards the M1-like anti-tumor phenotype to unleash their potential against tumors has become one of the most promising anti-tumor immunotherapy strategies. In this work, the natural product pseudolaric acid B (PAB, **1**) was found to markedly decrease ARG1 mRNA expression and significantly increase NOS2 expression in the IL-4/IL-13-pre-stimulated RAW 264.7 cells through cellular phenotype screening of a series of pseudolaric acid-related natural products, suggesting its potential to reprogram the pro-tumoral TAMs towards the M1-like phenotype against tumors. Further chemical modification of the carboxylic acid moiety of **1** led to a series of amide or pyranoside derivatives with ARG1- and NOS2-modulating activity. Among them, hydrazineyl amide **12** stands out as the most potent, without significant diminution in cell viability. It inhibited the M2-like polarized tumor-promoting phenotype of macrophages, as evidenced by a decrease in CD206 expression and an increase in CD86 expression in flow cytometry, as well as a decrease in ARG1 protein level in Western blot assays. In addition, **12** could reverse the suppression of Ki67+, IFN γ+, and granzyme B^+^ CD8^+^ T cell proliferation and activation induced by pro-tumoral macrophages. More importantly, it could reshape the tumor immune microenvironment and inhibit tumor growth in immunocompetent murine tumor models. Hsp90 was predicted to be a potential target of **12** by a target fishing software, which was further demonstrated by molecular docking. Collectively, the amide derivative **12** of PAB demonstrated promising anti-tumor TAM-reprogramming activity, which is worthy of further investigation.

## 1. Introduction

Tumor-associated macrophages (TAMs) that infiltrate tumor tissues are the most abundant immune cells in the tumor microenvironment (TME) [1,2,3]. In the early stages of tumor development, TAMs display a pro-inflammatory and M1-like polarization phenotype, which is capable of recognizing and eliminating tumor cells. However, as the tumor progresses, TAMs typically exhibit an anti-inflammatory M2 phenotype, which has been demonstrated to suppress the recruitment and function of cytotoxic T cells, promote angiogenesis, and affect multiple other aspects of tumor immunity. All of these together generate an immunosuppressive TME that finally promotes tumor growth and metastasis and mediates resistance to anticancer therapy [4,5,6,7,8]. Numerous efforts have been undertaken to develop TAM-targeted agents, with the aim of either depleting TAMs or reprogramming their behaviors through the inhibition of recruitment, activation of anti-tumoral polarization, and modulation of phagocytosis. Some of these agents have demonstrated promising therapeutic effects in monotherapy, as well as in combination with chemotherapy or immunotherapy, in clinical trials [9,10,11,12]. However, TAMs are highly heterogeneous, and their phenotypes and functions are plastic within the TME. Treating them as a whole or depleting TAMs often shows limited anti-tumor efficacy, together with undesired effects [11,13,14]. Thus, reprogramming the behaviors of TAMs to unleash their anti-tumor potential, thereby antagonizing the immunosuppressive TME and effectively activating cytotoxic CD8^+^ T cells, is increasingly recognized as a critical component of effective anti-tumor strategies. Therefore, the discovery of therapeutic agents with new chemotypes for repolarizing pro-tumoral and immunosuppressive M2-like TAMs towards the anti-tumor M1-like phenotype is highly needed.

Natural products with rich structural diversity and complexity have been demonstrated to be invaluable sources of lead compounds for drug discovery against various cancers and immune system disorders [15,16]. Pseudolaric acid B (PAB) is a compact tricyclic diterpene acid isolated from the root and trunk bark of Pseudolarix kaempferi Gordon (Pinaceae), also known as “Tu-Jin-Pi” in traditional Chinese herbal medicine, for the treatment of various eczema and fungal skin infections for centuries [17,18]. As one of the major bioactive components of this species, PAB was found to possess a broad spectrum of therapeutic effects, including anticancer, antimicrobial, anti-angiogenesis, immunomodulatory, and anti-inflammatory properties [18,19,20]. It is known that PAB exerts anti-tumor effects by inhibiting cancer cell proliferation, inducing cancer cell apoptosis or autophagy, inhibiting angiogenesis, etc. [19,21,22]. However, most reports on the anti-tumor effects of PAB focus primarily on the tumor cells themselves, while its immunomodulatory effects on the tumor microenvironment, particularly on tumor-associated immune cells, are seldom reported. Some studies showed that PAB regulates the inflammatory response in various inflammatory models. For example, in macrophages, PAB could inhibit the mRNA and protein levels of inflammatory cytokines, such as TNF-α, IL-6, etc., which are crucial factors in inflammation-driven tumorigenesis [23,24,25,26,27]. With its demonstrated efficacy in immunomodulation, PAB and its analogs might be used as lead compounds to pursue new derivatives to reprogram macrophages in cancer, enhancing the anti-tumor immune response. PAB is structurally characterized by a trans-fused polyhydroazulene core bearing a bridging lactone ring at the ring junction, with a conjugated diene side chain that might contribute to the reported biological activities. Currently, most structural modifications on PAB have mainly focused on the conjugated diene carboxylic acid side chain, with the aim of improving the anti-tumor activity of PAB and reducing its cytotoxicity to normal cells [28,29,30,31]. In this study, we identified PAB from a series of natural pseudolaric acids possessing the potential to repolarize pro-tumoral and immunosuppressive M2-like macrophages towards the anti-tumor M1-like phenotype and also designed and synthesized a series of amide derivatives through introducing alkyl amino fragments to obtain potent TAM-targeted modulators against tumor growth.

## 2. Results

### 2.1. The Effects of Natural Pseudolaric Acids and Their O-Pyranosides on the Tumor-Promoting Phenotype of the RAW264.7 Cell Line

We first investigated the impacts of a series of commercially available natural pseudolaric acids or their O-pyranosides (Figure 1) on the pro-tumoral phenotype of TAMs, which play a critical role in anti-tumor immune response modulation. When exhibiting a pro-tumor M2 phenotype, TAMs are characterized by high levels of arginase 1 (ARG1), which depletes L-arginine within the TME, thus effectively suppressing CD8^+^ T cell activation through arginine metabolism. In contrast, M1-biased macrophages combat tumor growth and bacterial infections by expressing inducible nitric oxide synthase 2 (NOS2), which catalyzes the production of nitric oxide (NO) [7,32,33]. Recognizing the pivotal roles of ARG1 and NOS2, we utilized these molecules as key biomarkers and effectors for assessing macrophage immune function. In our experiments, the mouse monocyte–macrophage leukemia cell line RAW 264.7 was induced with IL-4/IL-13 to acquire an M2-like pro-tumoral phenotype. Concurrently, the cells were treated with the pseudolaric acid-related natural products **1**–**7** at a concentration of 10 μM for 12 h. Additionally, PLX3397 was used as a positive control at a concentration of 0.5 μM. When compared to the untreated control, IL-4/IL-13 stimulation resulted in a marked increase in ARG1 mRNA expression and a significant decrease in NOS2 mRNA expression, confirming the M2-like pro-tumoral phenotype. As shown in Table 1 and Appendix A, pseudolaric acid B (**1**), its deacetoxyl analogue **2** (pseudolaric acid C), and β-D-glucoside **7** substantially attenuated the upregulation of ARG1 mRNA expression induced by IL-4/IL-13, with statistically significant effects. In contrast, bis-carboxylic acids analogs **4** (pseudolaric acid C2) and **6** (demethoxydeacetoxypseudolaric acid B) have no significant impact on ARG1 mRNA expression. Interestingly, pseudolaric acid A (**3**) substantially upregulated the mRNA expression of ARG1, while its β-D-glucopyranoside **5** dramatically decreased ARG1 expression. On the other hand, **1** and its β-D-glucoside **7** were observed to upregulate NOS2 mRNA expression levels, while the deacetoxyl- or methyl ester-hydrolyzed analogs **2**, **4**, and **6** displayed no significant stimulating effects on NOS2 mRNA expression. Compound **3** could substantially upregulate the mRNA expression of NOS2, while its glycoside **5** failed to significantly upregulate NOS2 expression. Taken together, both **1** and its glucoside **7** displayed particularly robust inhibitory effects against ARG1 mRNA expression and induced a significant increase in NOS2 mRNA expression, suggesting their capacity to alter the tumor-suppressive phenotype of RAW264.7 cells. Moreover, only a minimal reduction in cell viability was observed after 12 h of treatment with compounds **1** and **7** (Appendix A), ruling out the possibility that the observed inhibition of the pro-tumoral macrophage phenotype was due to compromised cell viability. Collectively, natural product **1** could be used as a lead compound, in which the carboxylic acid moiety would be optimized to pursue a more potent and drug-like compound with the potential to reprogram TAMs towards the anti-tumor M1-like phenotype.

### 2.2. Design and Synthesis of PAB Derivatives

Since O-glucosidation (**7**) at the carboxylic acid moiety of **1** is tolerable for its bioactivity, we designed a series of amide and O-pynaroside derivatives at this site. The synthetic method of the amide derivatives of **1** is outlined in Figure 1. The amidization of commercially available PAB with various alkyl amines under the condition of HATU/DIPEA/DMF gave rise to a series of compounds (**8**–**15**) with yields of 42–78%. As shown in Figure 2, the glycosylation of **1** with 2,3,4,6-tetra-O-acetyl-α-D-glucopyranosyl bromide, 2,3,4-tri-O-acetyl-α-D-xylopyranosyl bromide, or acetobromo-α-D-glucuronic acid methyl ester, respectively, under Ag_2_CO_3/_MeCN afforded the glycosylated products **16**–**18** at yields of 32–65%.

### 2.3. The Effects of the Carboxylic Acid-Modified Derivatives of PAB on the Tumor-Promoting Phenotype of the RAW264.7 Cell Line

With these carboxylic acid-modified derivatives in hand, we investigated their impact at 10 μM on ARG1 and NOS2 mRNA expression in RAW 264.7 cells pre-stimulated with IL-4/IL-13 to evaluate the effects on the pro-tumor phenotype of macrophages. As shown in Table 2 and Appendix A, most of the amide derivatives substantially suppressed the IL-4/IL-13-induced enhancement of ARG1 mRNA expression, with statistically significant effects. Among them, the hydroxyethyl hydrazineyl amide **12** displayed the most potent effect, both suppressing ARG1 mRNA expression and promoting NOS2, with relative expression rates of 1.4 ± 1.1% and 182.2 ± 5.3%, respectively, which are superior to that of lead compound **1**. Moreover, no significant diminution in cell viability was detected following 12 h of exposure to compound **12** (Appendix A), which excludes the likelihood that the inhibition of the pro-tumoral macrophage phenotype was a consequence of diminished cell viability. Hydroxyethyl amide **8** failed to display statistically significant modulating effects on ARG1 and NOS2 mRNA expression. The replacement of the hydroxyl group in compound **8** with a fluoro group (yielding compound **9**) significantly enhanced its inhibitory activity against ARG1 and its stimulatory effect on NOS2, with relative expression rates of 7.4% ± 7.9% and 148.7% ± 14.1%, respectively. The installation of a methyl (**10**) or hydroxymethyl (**11**) group at the hydroxyethyl amide moiety of **8** also exhibited statistically significant suppression activity against ARG1 expression, with relative expression rates of 7.5% and 17.5%, respectively, without significant impacts on NOS2 expression. The bis(2-hydroxyethyl) amide **13** did not significantly decrease ARG1 mRNA expression but showed a statistically significant stimulatory effect on NOS2. The glycerol amide **14** could significantly decrease ARG1 expression and increase the mRNA expression of NOS2. Further removal of one hydroxyl from the glycerol moiety **15** further enhanced the inhibitory effect on ARG1 expression but had no significant impact on NOS2 expression. Although glucoside **7** showed decent modulating effects on ARG1 and NOS2 mRNA expression, the further acetylated derivative **16** did not lose significant inhibitory or stimulatory effects. Both acetoxyl xylopyranoside **17** and acetoxyl glucuronic acid methyl ester **18** significantly decreased ARG1 mRNA expression, although they failed to upregulate NOS2 expression. Taken together, the hydrazineyl amide **12** demonstrated the best ability to reprogram pro-tumoral M2-like TAMs towards the M1-like anti-tumor phenotype, and was therefore selected for further in vitro and in vivo evaluation.

### 2.4. Compound ***12*** Inhibited the M2-like Polarized Tumor-Promoting Phenotype of Macrophages

We tested the inhibitory effects of compound **12** at multiple concentrations on the mRNA expression levels of ARG1, NOS2, the classic M2 marker CD206, and the classic M1 marker MHC II in RAW 264.7 cells stimulated with IL-4/IL-13 [34], and the results showed that compound **1** significantly reduced the mRNA expression levels of ARG1 and CD206. Concurrently, it upregulated the mRNA expression levels of NOS2 and MHC II in a dose-dependent manner (Figure 2A). We also extended the test to primary murine macrophages derived from bone marrow (BMDMs), which were cultured in vitro with recombinant mouse M-CSF for 6 days. The BMDMs were then stimulated with IL-4/IL-13, leading to their transformation into M2-like pro-tumoral macrophages. This transformation was characterized by an increase in both ARG1 and CD206 expression and a decrease in both NOS2 and MHC II expression. Notably, treatment with compound **12** significantly reversed these trends (Figure 2B), further substantiating its role in suppressing the M2 polarization of macrophages. Subsequently, we assessed the capacity of compound **12** to inhibit the M2-like polarization of macrophages using CSF-1-differentiated mouse BMDMs. These cells were treated with IL-4 and IL-13 for 48 h, and the expression levels of CD206, an M2-type marker, and CD86, an M1-type marker, were analyzed via flow cytometry. As anticipated, treatment with compound **12** effectively inhibited the M2-like polarization of macrophages, evidenced by a decrease in CD206 expression and an increase in CD86 expression in a dose-dependent fashion (Figure 2C). Moreover, compound **12** demonstrated potent inhibitory activity at the protein level for ARG1 expression, aligning with our mRNA-level experimental outcomes (Figure 2D). Taken together, these results indicated that compound **12** possesses the ability to reverse the M2-like polarized tumor-promoting phenotype of macrophages.

### 2.5. Compound ***12*** Reversed Macrophage-Mediated CD8^+^ T Cell Suppression

Pro-tumoral M2-like TAMs are well-known to directly suppress CD8^+^ T cell proliferation and activation [35,36]. When they were co-cultured with autologous spleen cells, as expected, the IL-4/IL-13-induced BMDMs suppressed the proliferation and activation of CD8^+^ T cells. Interestingly, pretreatment of BMDMs with compound **12** dose-dependently reversed such inhibitory effects on CD8^+^ T cell proliferation and activation, as evidenced by a significant increase in the proportion of Ki67^+^ CD8^+^ T cells, IFN γ^+^ CD8^+^ T cells, and GRZB^+^ CD8^+^ T cells in total CD8^+^ T cells (Figure 3). The above data indicated that compound **12** reversed the suppression of T cell proliferation and activation induced by pro-tumoral macrophages.

### 2.6. Pharmacokinetic Properties of Compound ***12*** in Mice

To investigate the druggability of compound **12**, the pharmacokinetic (PK) properties of **12** were examined in mice. The mice were treated with **12** intravenously (i.v.) at a dose of 1 mg/kg, and the resulting PK parameters were recorded. As shown in Table 3, compound **12** exhibited a half-life time (T_1/2_) of 40.2 h, a quick peak time of 0.083 h, and a peak plasma concentration (C_max_) of 32.1 ng/mL, together with a plasma exposure (AUC_0–∞_) of 363 h·ng/mL and high systemic plasma clearance of 2766 mL/hr/kg, indicating that the overall PK properties of MCB-20-341 are roughly suitable for further in vivo anticancer evaluation.

### 2.7. Compound ***12*** Reshaped the Tumor Immune Microenvironment and Inhibited Tumor Growth in Immunocompetent Murine Tumor Models

Encouraged by its intriguing cellular activities, we investigated the anti-tumor effect of compound **12** in vivo. The Hepa1-6 liver cancer model, an immunocompetent macrophage-rich subcutaneous murine tumor model, was thus established. In the Hepa1-6 xenograft model, intraperitoneal injections of compound **12** inhibited tumor growth with inhibitory rates of 53.2% and 48.3% at doses of 25 and 50 mg/kg, respectively (Figure 4A). Additionally, there was no obvious loss of body weight during the treatment, indicative of tolerance (Figure 4B). To further investigate whether the in vivo anti-tumor effect of compound **12** is associated with its inhibition of pro-tumoral macrophages and consequent activation of anti-tumor CD8^+^ T cells, we analyzed the infiltration of macrophages and CD8^+^ T cells in the tumor tissue of the Hepa1-6 model. Upon treatment with 25 mg/kg of compound **12**, there was an increasing trend in the infiltration of total immune cells (CD45^+^ cells) (Figure 4C), and the tumor infiltration of total TAMs, TAMs with the M2-liketumor-promoting phenotype (CD206^+^ TAMs), and immunosuppression TAMs (ARG1^+^ TAMs) in CD45^+^ cells was significantly reduced (Figure 4D–F), accompanied by the marginal upregulation of TAMs with the M1-like phenotype (CD86^+^ TAMs, MHC II^+^ TAMs) (Appendix A). In addition, the infiltration of CD8^+^ T cells and activated CD8^+^ T cells, including GRZB^+^ CD8^+^ T cells, IFNγ^+^ CD8^+^ T cells, and TNFα^+^ CD8^+^ T cells, in CD45^+^ cells was significantly upregulated (Figure 4G–J). These data suggest that compound **12** suppressed pro-tumoral macrophages and activated anti-tumor CD8^+^ T cells, thus exhibiting potent anti-tumor efficacy.

### 2.8. Molecular Docking of Compound ***12*** with Hsp90

To explore the potential action target of **12**, we first carried out target prediction using three classic software algorithms, PharmMapper (https://www.lilab-ecust.cn/pharmmapper/index.html), SuperPred (https://prediction.charite.de/), and NetInfer (https://lmmd.ecust.edu.cn/netinfer/), for reverse pharmacophore-based target fishing. It was found that Hsp90 was predicted to be a potential target of **12** by these target fishing algorithms. It has been reported that Hsp90 orchestrates M2 polarization of TAMs through the activation of oncogenic signaling cascades (MAPK, AKT, STAT3, NF-κB, and YAP1) [37,38,39] and stabilization of the immunomodulatory cytokine MIF (macrophage migration inhibitory factor) [40]. Targeting HSP90 may provide a therapeutic strategy to reverse the tumor-promoting phenotype of macrophages and impede tumors. Therefore, we inferred that HSP90 is an important potential target of **12**. Molecular docking was performed with Schrodinger (2019-1) to examine the binding mode and affinity of **12** with the HSP90 *N*-terminal domain. We obtained the top-ranked conformation of **12** with a docking score of −5.741 kcal/mol. Through the analysis of their binding mode (Figure 5), it was found that **12** could form several hydrogen bonds with multiple amino acid residues, such as Asp40, Asp42, Lys44, Asn92, and Thr171, on the HSP90 protein, further suggesting that derivative **12** might target HSP90.

## 3. Materials and Methods

### 3.1. Chemistry

All commercially available starting materials and solvents are of reagent grade and used without further purification. Column chromatography was performed using a 300–400-mesh or 200–300-mesh silica gel. Analytical TLC was carried out by employing 60 F254 plates, and spots were visualized using UV (254 or 365 nm). ^1^H NMR spectra were recorded with a AVANCE Ⅲ 400 MHz NMR spectrometer (Bruker, Billerica, MA, USA) and ^13^C NMR spectra were recorded with AVANCE NEO 700 MHz NMR spectrometer (Bruker, Billerica, MA, USA). Chemical shifts (δ) were reported in ppm downfield from an internal TMS standard, and *J* values were given in Hz. High-resolution mass spectra were obtained from an acquity UPLC/QTOF premier mass spectrometer (Waters, Milford, MA, USA). The purity of final compounds was determined using analytical HPLC, which was carried out on an Agilent Technologies 1260 series LC system with ultraviolet wavelengths in UV254 (Agilent Technologies, Inc., Santa Clara, CA, USA). HPLC analysis conditions: XDB-C18, 3.5 μm, 4.6 mm × 150 mm, and H_2_O/MeOH. All the assayed compounds displayed a purity of 95–100%.

General procedure for the synthesis of amide derivatives **8**–**15**.

To a solution of **1** (50 mg, 0.12 mmol) and alkyl amines (1.5 eq, 0.17 mmol) in DMF, HATU (66 mg, 0.17 mmol) and DIPEA (30 mg, 0.22 mmol) were added. The resulting mixture was stirred at rt overnight. The mixture was diluted with EtOAc, washed with brine, dried over Na_2_SO_4_, and concentrated in a vacuum. The resulting residue was purified using a silica gel column to give the desired amide products **8**–**15** at 42–78% yields.

Methyl(3R,4S,4aS,9aR)-4a-acetoxy-3-((1E,3E)-5-((2-hydroxyethyl)amino)-4-methyl-5-oxopenta-1,3-dien-1-yl)-3-methyl-1-oxo-3,4,4a,5,6,9-hexahydro-1H-4,9a-ethanocyclohepta[c]pyran-7-carboxylate (**8**, 35 mg, 64%). This compound was purified using a silica gel column (CH_2_Cl_2_/MeOH = 20:1; TLC: Rf = 0.2). ^1^H NMR (400 MHz, CDCl_3_) δ 7.18 (d, *J* = 7.9 Hz, 1H), 7.12–6.88 (m, 1H), 6.44 (ddd, *J* = 26.8, 15.0, 11.1 Hz, 1H), 6.16 (d, *J* = 10.9 Hz, 1H), 5.61 (d, *J* = 15.1 Hz, 1H), 3.84 (t, *J* = 4.8 Hz, 2H), 3.74–3.60 (m, 5H), 3.24 (d, *J* = 5.6 Hz, 1H), 3.04 (d, *J* = 14.2 Hz, 1H), 2.86 (d, *J* = 16.0 Hz, 1H), 2.71 (dd, *J* = 14.9, 8.8 Hz, 1H), 2.57 (d, *J* = 16.7 Hz, 1H), 2.10 (q, *J* = 2.4 Hz, 4H), 2.02–1.90 (m, 3H), 1.84–1.64 (m, 5H), 1.55 (d, *J* = 3.5 Hz, 4H). ^13^C NMR (176 MHz, CDCl_3_) δ 173.13, 169.80, 169.44, 168.10, 142.55, 141.77, 134.48, 133.17, 130.42, 121.38, 90.08, 83.84, 62.45, 55.23, 52.08, 49.31, 42.80, 33.31, 30.68, 28.54, 27.74, 24.28, 21.79, 20.12, 13.15. HRMS (ESI [M + Na]^+^) calcd for C_25_H_33_NNaO_8_, 498.2104; found, 498.2109.

Methyl (3R,4S,4aS,9aR)-4a-acetoxy-3-methyl-3-((1E,3E)-4-methyl-5-oxo-5-((2,2,2-trifluoroethyl)amino)penta-1,3-dien-1-yl)-1-oxo-3,4,4a,5,6,9-hexahydro-1H-4,9a-ethanocyclohepta[c]pyran-7-carboxylate (**9**, 26 mg, 44%). This compound was purified using a silica gel column (CH_2_Cl_2_/MeOH = 30:1; TLC: Rf = 0.2). ^1^H NMR (400 MHz, CDCl_3_) δ 7.16 (dt, *J* = 7.5, 3.0 Hz, 1H), 7.03–6.94 (m, 1H), 6.46 (dd, *J* = 15.0, 11.4 Hz, 1H), 6.32 (d, *J* = 14.3 Hz, 1H), 5.82 (d, *J* = 15.0 Hz, 1H), 4.06–3.75 (m, 2H), 3.68 (d, *J* = 2.5 Hz, 3H), 3.26 (d, *J* = 4.6 Hz, 1H), 3.04 (dd, *J* = 14.2, 6.3 Hz, 1H), 2.85 (dd, *J* = 15.6, 6.4 Hz, 1H), 2.79–2.64 (m, 1H), 2.63–2.53 (m, 1H), 2.17–2.04 (m, 4H), 1.98 (t, *J* = 1.1 Hz, 3H), 1.84–1.62 (m, 6H), 1.54 (s, 3H).^13^C NMR (176 MHz, CDCl_3_) δ 173.17, 169.42, 168.66, 168.04, 143.19, 141.70, 134.50, 134.09, 129.75, 121.15, 90.03, 83.89, 55.26, 52.06, 49.25, 40.97, 40.77, 33.32, 30.68, 28.50, 27.72, 24.31, 21.78, 20.11, 13.14. HRMS (ESI [M + Na]^+^) calcd for C_25_H_30_F_3_NNaO_7_, 536.1872; found, 536.1876.

Methyl(3R,4S,4aS,9aR)-4a-acetoxy-3-((1E,3E)-5-((2-hydroxypropyl)amino)-4-methyl-5-oxopenta-1,3-dien-1-yl)-3-methyl-1-oxo-3,4,4a,5,6,9-hexahydro-1H-4,9a-ethanocyclohepta[c]pyran-7-carboxylate (**10**, 25 mg, 45%). This compound was purified using a silica gel column (CH_2_Cl_2_/MeOH = 20:1; TLC: Rf = 0.1). ^1^H NMR (400 MHz, CDCl_3_) δ 7.15 (dt, *J* = 7.2, 3.0 Hz, 1H), 6.94 (d, *J* = 11.3 Hz, 1H), 6.52–6.34 (m, 2H), 5.79 (d, *J* = 15.0 Hz, 1H), 3.96–3.85 (m, 1H), 3.67 (s, 3H), 3.52–3.42 (m, 1H), 3.24 (d, *J* = 5.2 Hz, 1H), 3.16 (dddd, *J* = 13.3, 8.0, 5.1, 3.3 Hz, 1H), 3.07–2.98 (m, 1H), 2.84 (dd, *J* = 15.6, 6.3 Hz, 1H), 2.70 (dd, *J* = 15.1, 8.8 Hz, 1H), 2.57 (ddd, *J* = 15.0, 4.1, 1.9 Hz, 1H), 2.09 (s, 4H), 1.94 (d, *J* = 1.3 Hz, 3H), 1.82–1.64 (m, 6H), 1.54 (s, 3H), 1.16 (d, *J* = 6.3 Hz, 3H).^13^C NMR (176 MHz, CDCl_3_) δ 173.12, 169.70, 169.44, 168.10, 142.51, 141.77, 134.48, 133.13, 130.49, 121.40, 90.10, 83.84, 67.59, 55.24, 52.08, 49.32, 47.41, 33.31, 30.69, 28.55, 27.75, 24.30, 21.80, 21.04, 20.14, 13.18. HRMS (ESI [M + H]^+^) calcd for C_26_H_36_NO_8_, 490.2441; found, 490.2441.

Methyl(3R,4S,4aS,9aR)-4a-acetoxy-3-((1E,3E)-5-((2,3-dihydroxypropyl)amino)-4-methyl-5-oxopenta-1,3-dien-1-yl)-3-methyl-1-oxo-3,4,4a,5,6,9-hexahydro-1H-4,9a-ethanocyclohepta[c]pyran-7-carboxylate (**11**, 32 mg, 55%). This compound was purified using a silica gel column (CH_2_Cl_2_/MeOH = 15:1; TLC: Rf = 0.1). ^1^H NMR (400 MHz, CDCl_3_) δ 7.17 (d, *J* = 8.0 Hz, 1H), 6.98 (d, *J* = 11.4 Hz, 1H), 6.54 (s, 1H), 6.46 (dd, *J* = 15.0, 11.4 Hz, 1H), 5.83 (d, *J* = 15.0 Hz, 1H), 3.77 (*p*, *J* = 5.1 Hz, 1H), 3.69 (s, 3H), 3.53 (dd, *J* = 5.0, 2.2 Hz, 2H), 3.46 (ddd, *J* = 13.4, 9.8, 6.5 Hz, 1H), 3.26 (d, *J* = 4.8 Hz, 1H), 3.04 (dd, *J* = 14.1, 6.3 Hz, 1H), 2.85 (dd, *J* = 15.5, 6.3 Hz, 1H), 2.71 (dd, *J* = 15.1, 8.8 Hz, 1H), 2.63–2.53 (m, 1H), 2.10 (s, 3H), 1.95 (d, *J* = 1.3 Hz, 3H), 1.87–1.64 (m, 5H), 1.55 (s, 3H). ^13^C NMR (176 MHz, CDCl_3_) δ 173.21, 170.38, 169.45, 168.12, 142.94, 141.79, 134.50, 133.68, 129.94, 121.30, 90.05, 83.91, 71.16, 63.63, 55.24, 52.11, 49.29, 42.57, 33.33, 30.68, 28.52, 27.73, 24.31, 21.80, 20.12, 13.11. HRMS (ESI [M + Na]+) calcd for C_26_H_35_NNaO_9_, 528.2210; found, 528.2210.

Methyl(3R,4S,4aS,9aR)-4a-acetoxy-3-((1E,3E)-5-(2-(2-hydroxyethyl)hydrazineyl)-4-methyl-5-oxopenta-1,3-dien-1-yl)-3-methyl-1-oxo-3,4,4a,5,6,9-hexahydro-1H-4,9a-ethanocyclohepta[c]pyran-7-carboxylate (**12**, 24 mg, 42%). This compound was purified using a silica gel column (CH_2_Cl_2_/MeOH = 15:1; TLC: Rf = 0.2). ^1^H NMR (400 MHz, CDCl_3_) δ 7.16 (d, *J* = 4.6 Hz, 1H), 6.95 (d, *J* = 11.4 Hz, 1H), 6.52–6.35 (m, 2H), 5.79 (d, *J* = 15.1 Hz, 1H), 3.72 (t, *J* = 4.9 Hz, 2H), 3.68 (s, 3H), 3.47 (q, *J* = 5.2 Hz, 2H), 3.25 (d, *J* = 5.1 Hz, 1H), 3.09–2.99 (m, 1H), 2.85 (dd, *J* = 15.6, 6.3 Hz, 1H), 2.70 (dd, *J* = 15.1, 8.8 Hz, 1H), 2.57 (ddd, *J* = 15.1, 4.0, 1.8 Hz, 1H), 2.09 (s, 4H), 1.95 (d, *J* = 1.4 Hz, 3H), 1.86–1.63 (m, 5H), 1.54 (s, 3H). ^13^C NMR (176 MHz, CDCl_3_) δ 173.06, 169.44, 168.10,168.08, 141.79, 140.24, 134.45, 129.51, 121.00, 90.17, 83.77, 61.27,61.17, 55.20, 53.19, 52.06, 49.37, 33.30, 30.70, 28.66, 27.73, 24.28, 21.80, 20.12, 15.18. HRMS (ESI [M + H]^+^) calcd for C_25_H_35_N_2_O_8_, 491.2393; found, 491.2393.

Methyl(3R,4S,4aS,9aR)-4a-acetoxy-3-((1E,3E)-5-((2-(bis(2-hydroxyethyl)amino)ethyl)amino)-4-methyl-5-oxopenta-1,3-dien-1-yl)-3-methyl-1-oxo-3,4,4a,5,6,9-hexahydro-1H-4,9a-ethanocyclohepta[c]pyran-7-carboxylate (**13**, 32 mg, 50%). This compound was purified using a silica gel column (CH_2_Cl_2_/MeOH = 20:1; TLC: Rf = 0.1). ^1^H NMR (400 MHz, CDCl_3_) δ 7.16 (d, *J* = 8.2 Hz, 1H), 6.94 (d, *J* = 11.1 Hz, 2H), 6.44 (dd, *J* = 15.0, 11.3 Hz, 1H), 5.80 (d, *J* = 15.0 Hz, 1H), 3.68 (s, 3H), 3.60 (t, *J* = 4.9 Hz, 4H), 3.42 (q, *J* = 5.6 Hz, 2H), 3.24 (d, *J* = 5.3 Hz, 1H), 3.04 (dd, *J* = 14.0, 6.4 Hz, 1H), 2.93–2.80 (m, 1H), 2.74–2.65 (m, 7H), 2.57 (ddd, *J* = 15.1, 4.2, 1.7 Hz, 1H), 2.09 (s, 4H), 1.92 (d, *J* = 1.3 Hz, 3H), 1.83–1.64 (m, 5H), 1.54 (s, 3H). ^13^C NMR (176 MHz, CDCl_3_) δ 173.31, 169.75, 169.44, 168.10, 142.28, 141.83, 134.43, 132.89, 130.81, 121.51, 90.02, 83.98, 59.15, 56.63, 55.20, 54.67, 52.06, 49.32, 38.12, 33.30, 30.65, 28.48, 27.71, 24.25, 21.77, 20.10, 12.96. HRMS (ESI [M + H]^+^) calcd for C_29_H_43_N_2_O_9_, 563.2969; found, 563.2966.

Methyl(3R,4S,4aS,9aR)-4a-acetoxy-3-((1E,3E)-5-((1,3-dihydroxy-2-(hydroxymethyl)propan-2-yl)amino)-4-methyl-5-oxopenta-1,3-dien-1-yl)-3-methyl-1-oxo-3,4,4a,5,6,9-hexahydro-1H-4,9a-ethanocyclohepta[c]pyran-7-carboxylate (**14**, 47 mg, 78%). This compound was purified using a silica gel column (CH_2_Cl_2_/MeOH = 12:1; TLC: Rf = 0.1). ^1^H NMR (400 MHz, CDCl_3_) δ 7.20–7.11 (m, 1H), 6.94–6.84 (m, 2H), 6.46 (ddd, *J* = 13.1, 11.3, 2.8 Hz, 1H), 5.83 (d, *J* = 14.9 Hz, 1H), 5.19 (s, 3H), 3.68 (d, *J* = 1.5 Hz, 2H), 3.57 (d, *J* = 5.5 Hz, 6H), 3.26 (d, *J* = 5.0 Hz, 1H), 3.04 (dd, *J* = 14.2, 6.5 Hz, 1H), 2.96–2.79 (m, 1H), 2.77–2.66 (m, 2H), 2.62–2.53 (m, 1H), 2.16–2.04 (m, 3H), 2.00 (d, *J* = 6.1 Hz, 2H), 1.94 (s, 2H), 1.84–1.65 (m, 5H), 1.55 (d, *J* = 1.7 Hz, 3H).^13^C NMR (176 MHz, CDCl_3_) δ 173.05, 170.03, 169.46, 168.10, 143.21, 141.74, 134.49, 133.62, 130.45, 121.31, 90.10, 83.80, 61.89, 61.75, 61.40, 55.24, 52.09, 49.28, 38.63, 33.32, 30.69, 28.53, 27.75, 24.33, 21.81, 20.14, 13.24. HRMS (ESI [M + H]^+^) calcd for C_27_H_38_NO_10_, 536.2496; found, 536.2498.

(2E,4E)-5-((3R,4S,4aS,9aR)-4a-acetoxy-7-(methoxycarbonyl)-3-methyl-1-oxo-3,4,4a,5,6,9-hexahydro-1H-4,9a-ethanocyclohepta[c]pyran-3-yl)-2-methylpenta-2,4-dienoic acid (**15**, 39 mg, 65%). This compound was purified using a silica gel column (CH_2_Cl_2_/MeOH = 20:1; TLC: Rf = 0.1). ^1^H NMR (400 MHz, CDCl_3_) δ 7.17 (dd, *J* = 7.8, 4.3 Hz, 1H), 6.91 (d, *J* = 11.3 Hz, 1H), 6.45 (dd, *J* = 15.0, 11.4 Hz, 1H), 6.33 (d, *J* = 3.3 Hz, 1H), 5.81 (d, *J* = 15.0 Hz, 1H), 4.19 (s, 1H), 3.68 (d, *J* = 1.8 Hz, 4H), 3.60 (d, *J* = 11.5 Hz, 2H), 3.25 (d, *J* = 5.0 Hz, 1H), 3.09–2.99 (m, 1H), 2.85 (dd, *J* = 15.6, 6.3 Hz, 1H), 2.70 (dd, *J* = 15.1, 8.8 Hz, 1H), 2.57 (ddd, *J* = 15.0, 4.1, 1.9 Hz, 1H), 2.09 (s, 4H), 2.00–1.88 (m, 4H), 1.83–1.64 (m, 4H), 1.54 (s, 3H), 1.24 (s, 3H), 0.96–0.76 (m, 2H).^13^C NMR (176 MHz, CDCl_3_) δ 173.05, 169.72, 169.44, 168.09, 142.88, 141.73, 134.50, 133.26, 130.82, 121.34, 90.10, 83.78, 67.53, 58.98, 55.24, 52.08, 49.30, 33.31, 30.69, 28.55, 27.76, 24.31, 21.81, 20.24, 20.22, 20.14, 13.33. HRMS (ESI [M + H]^+^) calcd for C_27_H_38_NO_9_,520.2547; found, 520.2549.

General procedure for the synthesis of amide derivatives **16**–**18**.

To a solution of **1** (70 mg, 0.16 mmol) and bromopyranose (1.0 eq, 0.15 mmol) in MeCN, Ag_2_CO_3_ (81 mg, 0.30 mmol) was added. The resulting mixture was stirred at rt for 4 h. The mixture was diluted with EtOAc, washed with brine, dried over Na_2_SO_4_, and concentrated in a vacuum. The resulting residue was purified using a silica gel column to give the desired pyranosides **16**–**18** at 32–65% yields.

(3R,5R,6R)-2-(((2E,4E)-5-((3R,4S,4aS,9aR)-4a-Acetoxy-7-(methoxycarbonyl)-3-methyl-1-oxo-3,4,4a,5,6,9-hexahydro-1H-4,9a-ethanocyclohepta[c]pyran-3-yl)-2-methylpenta-2,4-dienoyl)oxy)-6-(acetoxymethyl)tetrahydro-2H-pyran-3,4,5-triyl triacetate (**16**, 28 mg, 32%). This compound was purified using a silica gel column (CH_2_Cl_2_/MeOH = 25:1; TLC: Rf = 0.1). ^1^H NMR (400 MHz, CDCl_3_) δ 7.19 (s, 1H), 6.52 (dd, *J* = 15.1, 11.5 Hz, 1H), 5.92 (d, *J* = 15.1 Hz, 1H), 5.73 (d, *J* = 8.4 Hz, 1H), 5.45–5.35 (m, 2H), 5.09 (dd, *J* = 10.4, 3.3 Hz, 1H), 4.11 (td, *J* = 17.8, 17.3, 8.5 Hz, 3H), 3.69 (d, *J* = 1.9 Hz, 3H), 3.28 (s, 1H), 3.06 (dd, *J* = 14.5, 6.2 Hz, 1H), 2.89 (dd, *J* = 17.8, 11.7 Hz, 1H), 2.72 (dd, *J* = 15.1, 8.8 Hz, 1H), 2.59 (d, *J* = 14.9 Hz, 1H), 2.15 (s, 3H), 2.11 (s, 4H), 2.02 (d, *J* = 1.8 Hz, 3H), 1.99 (d, *J* = 3.4 Hz, 6H), 1.93 (s, 3H), 1.85–1.66 (m, 4H), 1.59 (s, 3H).^13^C NMR (176 MHz, CDCl_3_) δ 172.85, 170.40, 170.16, 170.00, 169.52, 169.42, 168.05, 165.90, 145.06, 141.68, 139.26, 134.48, 126.74, 121.53, 92.55, 90.06, 83.69, 71.70, 70.76, 67.74, 66.83, 60.98, 55.24, 52.07, 49.18, 33.28, 30.68, 29.06, 28.44, 27.72, 24.32, 21.80, 20.72, 20.68, 20.59, 20.12, 12.79. HRMS (ESI [M + Na]^+^) calcd for C_37_H_46_NaO_17_, 785.2633; found, 785.2633.

(2R,3R,4S,5R)-2-(((2E,4E)-5-((3R,4S,4aS,9aR)-4a-Acetoxy-7-(methoxycarbonyl)-3-methyl-1-oxo-3,4,4a,5,6,9-hexahydro-1H-4,9a-ethanocyclohepta[c]pyran-3-yl)-2-methylpenta-2,4-dienoyl)oxy)tetrahydro-2H-pyran-3,4,5-triyl triacetate (**17**, 35 mg, 44%). This compound was purified using a silica gel column (CH_2_Cl_2_/MeOH = 25:1; TLC: Rf = 0.1). ^1^H NMR (400 MHz, CDCl_3_) δ 7.18 (d, *J* = 10.7 Hz, 2H), 6.52 (dd, *J* = 15.1, 11.4 Hz, 1H), 5.90 (d, *J* = 15.0 Hz, 1H), 5.76 (d, *J* = 6.8 Hz, 1H), 5.22 (td, *J* = 8.2, 0.0 Hz, 1H), 5.09 (dd, *J* = 8.3, 6.8 Hz, 1H), 4.97 (td, *J* = 8.3, 5.0 Hz, 1H), 4.14 (dd, *J* = 12.0, 4.9 Hz, 1H), 3.70 (s, 3H), 3.54 (dd, *J* = 12.1, 8.4 Hz, 1H), 3.29 (d, *J* = 3.4 Hz, 1H), 3.06 (dd, *J* = 14.2, 6.3 Hz, 1H), 2.87 (dd, *J* = 15.5, 6.3 Hz, 1H), 2.72 (dd, *J* = 15.1, 8.8 Hz, 1H), 2.59 (d, *J* = 14.7 Hz, 1H), 2.11 (s, 3H), 2.05 (d, *J* = 0.9 Hz, 5H), 2.02 (s, 3H), 1.93 (d, *J* = 1.4 Hz, 3H), 1.86–1.66 (m, 4H), 1.57 (s, 6H).^13^C NMR (176 MHz, CDCl_3_) δ 172.84, 169.88, 169.77, 169.41, 168.04, 166.00, 144.96, 141.68, 138.97, 134.48, 127.03, 121.52, 92.35, 90.06, 83.67, 70.83, 69.41, 68.29, 62.73, 55.24, 52.06, 49.17, 33.29, 30.68, 28.44, 27.73, 24.34, 21.80, 20.76, 20.72, 20.67, 20.12, 12.78. HRMS·(ESI [M + Na]^+^) calcd for C_34_H_42_NaO_15_, 713.2421; found, 713.2427.

(2R,3S,4S,5R,6R)-6-(((2E,4E)-5-((3R,4S,4aS,9aR)-4a-Acetoxy-7-(methoxycarbonyl)-3-methyl-1-oxo-3,4,4a,5,6,9-hexahydro-1H-4,9a-ethanocyclohepta[c]pyran-3-yl)-2-methylpenta-2,4-dienoyl)oxy)tetrahydro-2H-pyran-2,3,4,5-tetrayl tetraacetate (**18**, 56 mg, 65%). This compound was purified using a silica gel column (CH_2_Cl_2_/MeOH = 15:1; TLC: Rf = 0.2). ^1^H NMR (400 MHz, CDCl_3_) δ 7.19 (d, *J* = 11.3 Hz, 2H), 6.50 (dd, *J* = 15.0, 11.4 Hz, 1H), 5.92 (d, *J* = 15.0 Hz, 1H), 5.80 (d, *J* = 7.9 Hz, 1H), 5.33 (t, *J* = 9.2 Hz, 1H), 5.22 (td, *J* = 9.2, 3.0 Hz, 2H), 4.18 (d, *J* = 9.7 Hz, 1H), 3.69 (d, *J* = 2.3 Hz, 6H), 3.29 (d, *J* = 3.4 Hz, 1H), 3.06 (dd, *J* = 14.3, 6.3 Hz, 1H), 2.87 (dd, *J* = 15.6, 6.3 Hz, 1H), 2.72 (dd, *J* = 15.1, 8.8 Hz, 1H), 2.63–2.53 (m, 1H), 2.11 (s, 3H), 2.01 (d, *J* = 11.4 Hz, 8H), 1.91 (d, *J* = 1.4 Hz, 3H), 1.85–1.65 (m, 5H), 1.63–1.60 (m, 2H), 1.57 (s, 3H). ^13^C NMR (176 MHz, CDCl_3_) δ 172.85, 169.48, 169.40, 169.27, 168.06, 166.83, 165.71, 145.25, 141.69, 139.55, 134.48, 126.54, 121.44, 91.72, 90.04, 83.67, 72.94, 71.70, 69.95, 69.08, 55.24, 53.02, 52.06, 49.17, 33.28, 30.67, 28.42, 27.74, 24.32, 21.80, 20.71, 20.60, 20.51, 20.12, 12.74. HRMS·(ESI [M + Na]+) calcd for C_37_H_46_NaO_17_, 771.2476; found, 771.2476.

### 3.2. Cell Culture

RAW 264.7 and Hepa1-6 cells were obtained from the American Type Culture Collection (Manassas, VA, USA). All cell lines in this study were maintained in the appropriate medium as suggested by suppliers and were authenticated via single-nucleotide polymorphism (SNP) analysis, with the latest test in May 2024 (Crown Bioscience, Taicang, China).

### 3.3. Macrophage Culture and Stimulation

The protocols for animal handling were approved by the Institutional Animal Care and Use Committee at Shanghai Institute of Materia Medica and performed according to the institutional ethical guidelines on animal care (approval no. 2024-06-GMY-38). Bone marrow cells were isolated from the tibia and femur of 6–8-week-old female C57BL/6 mice, seeded at a density of 2 × 10^6^ cells/well in a 6-well plate, and differentiated in the presence of M-CSF (20 ng/mL) and 10% fetal bovine serum in IMDM growth medium for 6 days. The medium was changed every three days. To fully polarize M2 macrophages, macrophages were stimulated with 20 ng/mL IL-4/IL-13. In certain experiments, macrophages were treated with different concentrations of compounds.

### 3.4. Quantitative Real-Time PCR (RT-PCR)

Total RNA was isolated from cells using the EZ-press RNA Purification Kit (EZBioscience, Suzhou, China) and subjected to reverse transcription with the 5×HiScript II qRT SuperMix II (Vazyme, Nanjing, China). PCR was performed with the 2×ChamQ Universal SYBR qPCR Master Mix (Vazyme, Nanjing, China). All primers for qRT-PCR are described in Appendix A. The 2^−∆∆Ct^ method was utilized to calculate the fold change in gene expression, with the expression levels in the stimulated group set as the baseline for normalization.

### 3.5. Macrophage Phenotype Analysis

BMDMs were collected and stained with a fluorescent antibody or the matching isotype controls for 30 min at room temperature and then tested using a BD CytoFLEX LX (Beckman, CA, USA). Antibodies specific for the following proteins and the matching isotype control or FMO control were used to analyze the macrophage’s phenotype: CD11b, F4/80, CD206, and CD86 (BD, Franklin Lakes, NJ, USA; eBioscience, San Diego, CA, USA; Biolegend, San Diego, CA, USA). Data were analyzed using FlowJo10.4 software. The gating strategies for flow cytometry are shown in Appendix A.

### 3.6. Western Blot Analysis

Total cellular protein was extracted using a 1 × SDS lysis buffer and denatured at 100 °C for 15 min. Then, the protein samples were loaded in 10% SDS-PAGE and transferred to a nitrocellulose membrane. After 1 h of blocking with 3% BSA (Sigma-Aldrich, St. Louis, MO, USA) at room temperature, the membrane was incubated with the primary antibodies ARG1 (#93668; 1:20,000) from Cell Signaling Technology (Danvers, MA, USA) and GAPDH (#KC-5G4; 1:20,000) from Kangcheng Bio (Shanghai, China) at 4 °C overnight, respectively. The membrane was then incubated with the corresponding secondary antibodies from Jackson Immuno Research (West Grove, PA, USA) (HRP-conjugated anti-rabbit IgG (#11-035-003; 1:2000) and HRP-conjugated anti-mouse IgG (#115-035-003; 1:2000)) at room temperature for 1 h, respectively. The Clarity Western ECL Substrate (Bio-Rad, Hercules, CA, USA) was used to visualize the blots, and images were captured using an ImageQuant LAS-4000 imager (GE Healthcare, Chicago, IL, USA).

### 3.7. Cell Proliferation Assay

Cells were seeded in 96-well tissue culture plates. On the next day, cells were exposed to various concentrations of compounds and further cultured for the indicated period. Finally, cell proliferation was determined by using the Cell Counting Kit (CCK-8) (Dojindo, Kumamoto, Japan) assay.

### 3.8. CD8^+^ T Cell Suppression Assay

Spleen cells were isolated from C57BL/6 mice, followed by red blood cell (RBC) lysis. BMDMs were induced into M2-like macrophages and treated with different concentrations of compound 1 for 48 h. Then, 2 × 10^5^ spleen cells/well were stimulated with αCD3/αCD28 and co-cultured with 2 × 10^4^ pro-treated BMDMs in 96-well plates in LCM (RPMI 1640 with 50 mM 2-mercaptoethanol and 10% fetal bovine serum) at 37 °C. After 48 h, the spleen cells were treated with the eBioscience™ Cell Stimulation Cocktail (plus protein transport inhibitors) for 4 h, and cell activation was then determined by the proportion of IFNγ^+^ CD8^+^ T cells and GRZB^+^ CD8^+^ T cells in CD8^+^ T cells measured using flow cytometry. Cell proliferation was determined by the proportion of Ki67^+^ CD8^+^ T cells in CD8^+^ T cells using flow cytometry. The gating strategies for flow cytometry are shown in Appendix A.

### 3.9. In Vivo Pharmacokinetic Studies in Mice

ICR mice (*n* = 3) were administered with the test compound via intravenous injection at 1 mg/kg. Blood samples were collected 0.083, 0.25, 0.5, 1, 2, 4, 8, and 24 h after administration. An aliquot of 10 μL of the plasma sample was deproteinized with 200 μL of a MeOH: ACN = 1: 1 solution containing an internal standard. After centrifugation, the supernatant was injected for LC–MS/MS analysis. A standard set of parameters, including the area under the curve (AUC_0–∞_), elimination half-life (T_1/2_), time to reach the maximum plasma concentration (T_max_), maximum plasma concentration (C_max_), plasma clearance (Cl), and mean residence time (MRT), were calculated using WinNonlin software.

### 3.10. In Vivo Anti-Tumor Efficacy

Animal procedures were approved by the Institutional Animal Care and Use Committee of the Shanghai Institute of Materia Medica (approval no. 2024-06-DJ-95). Mice (4–6 weeks old) were housed with five or six mice per cage in a specific pathogen-free room with a 12 h light/dark schedule at 25 °C ± 1 °C and were fed an autoclaved chow diet and water ad libitum. Hepa1-6 cells (2 × 10^6^ cells) were subcutaneously implanted in the right flank of C57 BL/6 mice. When the tumors reached a volume of around 50 mm^3^, the mice were randomized into the treatment groups. The vehicle groups were given the vehicle alone, and the treatment groups received compound **1** in the indicated doses via intratumoral injections once daily on the indicated days. The tumor volumes and body weights were measured twice per week. Tumor volume (TV) was calculated as follows: TV = (length × width^2^)/2. Significant differences between the treated versus the vehicle groups were determined using the Student’s *t*-test. For analyses of tumor-infiltrating immune cells, as depicted in the panels of Figure 4C–J, it is noted that in the 25 mg/kg treatment group, one mouse exhibited no evidence of a tumor. Consequently, the sample size for this treatment group was reduced to 7. Hepa1-6 tumor tissues were minced and digested using a Mouse Tumor Dissociation Kit (Miltenyi, Bergisch Gladbach, Germany). The cells were passed through a 70 μm cell strainer, stained with a fluorescent antibody or the matching isotype controls for 30 min at room temperature, and then tested using a BD CytoFLEX LX (Beckman). Antibodies specific for the following proteins and the matching isotype control or FMO control were used to analyze the leukocyte infiltrate: CD45, CD11b, F4/80, CD206, CD86, ARG1, MHCII, CD3e, CD8a, IFNγ, TNFα, and GRZB (BD, eBioscience, and Biolegend). Viability was determined by staining with either the LIVE/DEAD^®^ Fixable Violet Dead Cell Stain Kit (Invitrogen, Waltham, CA, USA) or the Zombie Aqua™ Fixable Viability Kit (Biolegend, San Diego, CA, USA). Data were analyzed using FlowJo10.4 software. The gating strategies for flow cytometry are shown in Appendix A.

### 3.11. Molecular Docking

The wild-type structure 7K9U with low resolution from the PDB database was selected for docking. In the preparation of the protein, the crystal structure was treated with the Protein Preparation Wizard module in the Schrodinger/2018 simulation package with the default settings, including removing redundant chains and water molecules, repairing the missing loops of the proteins and side chains of the residues, adding hydrogen atoms, and minimizing the systems. Compound **12** was prepared with the LigPrep module in Schrodinger/2018 with the default parameters, including determining the protonated states (pH = 7.0 ± 2.0), the tautomers, the stereoisomers, and so forth. The standard precision of Glide docking (Glide SP) in Schrödinger/2018 was used to generate the initial protein–ligand complex for HSP90 and 12. The receptor grid was set to 20 Å × 20 Å × 20 Å centered on the co-crystallized ligand of the system. The top-ranked docking pose of each ligand was kept for further analysis.

### 3.12. Statistical Analysis

Statistical analysis in this paper was conducted using GraphPad Prism 9 software (version 9.0.0; GraphPad Software, La Jolla, CA, USA).

## 4. Conclusions

TAMs within the TME are pivotal for tumor development and progression. Reprogramming the M2-like pro-tumoral behavior of TAMs towards the M1-like anti-tumor phenotype to unleash their potential against tumors has become one of the most promising anti-tumor immunotherapies. Although some of these agents have demonstrated promising therapeutic effects in clinical trials, TAM-reprogramming immunotherapies still suffer from limited anti-tumor efficacy, together with undesired effects, due to the highly heterogeneous and plastic characteristics of TAMs. Therefore, the discovery of novel chemotype small molecules with the therapeutic potential to repolarize pro-tumoral and immunosuppressive M2-like tumor-associated macrophages (TAMs) towards an anti-tumor phenotype is both highly necessary and extremely challenging. In this work, through cellular phenotype screening of a series of pseudolaric acid-related natural products, it was found that treating IL-4/IL-13-pre-stimulated RAW 264.7 cells with natural product PAB (**1**) or its glucoside 7 could induce a marked decrease in ARG1 mRNA expression and a significant increase in NOS2 expression, confirming their potential to reprogram the pro-tumoral TAMs towards the M1-like phenotype against tumors. Further chemical modification of the carboxylic acid moiety of **1** led to a series of amide or pyranoside derivatives with ARG1- and NOS2-modulating activity. Among them, hydrazineyl amide **12** stands out as the most potent, without significant diminution of cell viability of macrophages. It inhibited the M2-like polarized tumor-promoting phenotype of macrophages, as evidenced by a decrease in CD206 expression and an increase in CD86 expression in flow cytometry, as well as a decrease in ARG1 protein levels in Western blot assays. Moreover, 12 could reverse the suppression of Ki67, IFNγ, and granzyme B expression in CD8^+^ T cells, as well as the inhibition of the proliferation and activation of CD8^+^ T cells, which are induced by pro-tumoral immunosuppressive macrophages. The overall PK properties of **12** are basically suitable for in vivo evaluation. More importantly, it can reshape the tumor immune microenvironment and inhibit tumor growth in immunocompetent murine tumor models. Hsp90 was predicted to be a potential target of **12** by target fishing software, which was further demonstrated by molecular docking. Further target verification will be carried out in the near future. Our success in identifying new amide derivatives of PAB for the repolarization of pro-tumoral and immunosuppressive TAMs towards the anti-tumor phenotype will provide new chemotype lead compounds for developing TAM-reprogramming immunotherapies for cancer.

Additionally, it is widely acknowledged that chronic inflammation has been firmly established as one of the primary drivers in the initiation and progression of tumors [41,42,43]. Macrophages play a crucial role in both the initiation and resolution of inflammation. Previous studies showed that PAB inhibited the production of inflammatory cytokines in macrophages, including IL-1β, TNF-α, IL-6, etc. [23,24,25,26,27], which are crucial factors in inflammation-driven tumorigenesis. Thus, in addition to its well-established direct cytotoxic effect on tumor cells, PAB and its analogs could exert a synergistic anti-tumor effect by inhibiting pro-tumor inflammation and reversing the pro-tumor phenotype of tumor-associated macrophages. All these observations suggest the potential application of PAB and its analogs in the treatment of inflammation-driven or macrophage-rich cancers. It should be noted that several proteins, including CD147, Smoothened (SMO), and Hsp90, have been reported as the direct targets of the natural product PAB in tumor cells [44,45,46], as well as the direct target of **12.** Reprogramming TAM activity against tumors remains to be further elucidated.

## Data Availability

All data included in this study are available upon request by contacting the corresponding author.

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
