# Peer review of "Discovery of Hydrazineyl Amide Derivative of Pseudolaric Acid B for Reprogramming Tumor-Associated Macrophages Against Tumor Growth"

_molecules, 2025, doi:10.3390/molecules30102088_

Round 1
Reviewer 1 Report
Comments and Suggestions for Authors
This manuscript describes the use of pseudolaric acid B derivatives to reprogram the M2-like Tumor-Associated Macrophages (TAMs) towards the M1-like phenotype. The M2 phenotype suppresses the recruitment and function of cytotoxic T cells, which facilitates tumor growth, while the M1 phenotype can recognize and eliminate tumor cells. Methods to revert TAMS to the M1 phenotype are therefore of interest as cancer therapies. The researchers discover that pseudolaric acid derivatives are effective at modifying the phenotype of these TAMs, and a small number of analogs is generated and evaluated. One compound, the hydrazine 12, is especially promising, and it is evaluated further. Overall the manuscript is well crafted, and the story will be of interest to many organic and medicinal chemists. I did find a few minor points that should be addressed, including the following:
-Some sentences seem to be incomplete, like page 8 (line 353), page 10 (line 409), and page 11 (line 489) page 12 (line 502), and page 12 (line 509)
-The numbering becomes odd on page 10, as a second numbering system seems to be used (perhaps from a dissertation?). It would be helpful to the reader to only use one numbering system.
-The yield for each new compound should be included with the compound (instead of the range for all the compounds that is currently reported), as well as the amount prepared (for example, 65%, 210 mg).
-The solvent system used in the silica gel chromatography for each compound should be listed in the experimental section, as should the TLC Rf
-Infrared spectral data is needed for new compounds, no IR data is provided
-For the provided NMR spectra in the SI, the solvent used and field strength of the instrument should be included in the titles under the spectra
Author Response
Our point-by-point response is summarized below:
Reviewer 1
This manuscript describes the use of pseudolaric acid B derivatives to reprogram the M2-like Tumor-Associated Macrophages (TAMs) towards the M1-like phenotype. The M2 phenotype suppresses the recruitment and function of cytotoxic T cells, which facilitates tumor growth, while the M1 phenotype can recognize and eliminate tumor cells. Methods to revert TAMS to the M1 phenotype are therefore of interest as cancer therapies. The researchers discover that pseudolaric acid derivatives are effective at modifying the phenotype of these TAMs, and a small number of analogs is generated and evaluated. One compound, the hydrazine 12, is especially promising, and it is evaluated further. Overall the manuscript is well crafted, and the story will be of interest to many organic and medicinal chemists. I did find a few minor points that should be addressed, including the following:
Response: We are thankful for the positive comments from the reviewer.
-Some sentences seem to be incomplete, like page 8 (line 353), page 10 (line 409), and page 11 (line 489) page 12 (line 502), and page 12 (line 509)
Response: We thank the reviewer for the advice. As requested, we have carefully revised these sentences to ensure that they are grammatically correct and clear.
-The numbering becomes odd on page 10, as a second numbering system seems to be used (perhaps from a dissertation?). It would be helpful to the reader to only use one numbering system.
Response: We thank the reviewer for the advice. As requested, we altered the style of numbering and displayed them in a consistent form in the revised manuscript (Page 11, Line 446-453).
-The yield for each new compound should be included with the compound (instead of the range for all the compounds that is currently reported), as well as the amount prepared (for example, 65%, 210 mg).
Response: We appreciate the reviewer’s suggestion, and have added the yields and amount for each newly prepared compound in the revised manuscript.
-The solvent system used in the silica gel chromatography for each compound should be listed in the experimental section, as should the TLC Rf
Response: We thank the reviewer for the valuable suggestion. We have added the solvent system and TLC Rf value in the revised manuscript.
-Infrared spectral data is needed for new compounds, no IR data is provided
Response: We thank the reviewer for pointing this out. The new compounds in our manuscript were fully characterized by 1H, 13C-NMR spectra and HRMS together with HPLC data for demonstrating the purity. We think that these characterizations are enough for identifying the new compounds. Infrared spectral data is also very useful for determining the compound structure. Since there was no Infrared spectral instrument in our lab, we did not include IR data in the manuscript.
-For the provided NMR spectra in the SI, the solvent used and field strength of the instrument should be included in the titles under the spectra
Response: We thank the reviewer for this suggestion. We have the solvent used and field strength of the instrument in the title of the spectra.

Reviewer 2 Report
Comments and Suggestions for Authors
This manuscript presents a comprehensive study on the discovery and development of hydrazineyl amide derivatives of pseudolaric acid B (PAB) for reprogramming tumor-associated macrophages (TAMs) against tumor growth. Through systematic chemical modification and biological evaluation, the authors have developed a series of derivatives, among which compound 12 showed the most promising activity. The manuscript is well organized and written well. The manuscript can be accepted by Molecules after addressing the following issues:
- Although the authors have demonstrated the effect of compound 12 on TAMs, further mechanistic studies are needed to understand the precise molecular targets and signaling pathways involved. The authors are suggested to use at least reverse docking and possibly target fishing to achieve it.
- While the in vivo efficacy in the Hepa1-6 model is promising, additional tumor models should be considered to assess the broad applicability and efficacy of compound 12.
- Information on the PK/PD and toxicity profiles of compound 12 is lacking. These studies are crucial for the further development of the compound as a therapeutic agent.
- Either use CDCl3 or Chloroform-d. CNMR was done at a 700 MHz NMR spectrometer? (as it is shown as 176 MHz)
Author Response
Our point-by-point response is summarized below:
Reviewer 2
This manuscript presents a comprehensive study on the discovery and development of hydrazineyl amide derivatives of pseudolaric acid B (PAB) for reprogramming tumor-associated macrophages (TAMs) against tumor growth. Through systematic chemical modification and biological evaluation, the authors have developed a series of derivatives, among which compound 12 showed the most promising activity. The manuscript is well organized and written well. The manuscript can be accepted by Molecules after addressing the following issues:
Response: We are thankful for the positive comments from the reviewer.
- Although the authors have demonstrated the effect of compound 12 on TAMs, further mechanistic studies are needed to understand the precise molecular targets and signaling pathways involved. The authors are suggested to use at least reverse docking and possibly target fishing to achieve it.
Response: We thank the reviewer for pointing this out. To explore the potential action target of 12, we first carried out the target prediction using three classic software algorithms including PharmMapper, SuperPred, and NetInfer for reverse pharmacophore-based target fishing. It was found that Hsp90 was predicted to be potential target of 12 by these target fishing algorithms.
Actually, it has been reported that Hsp90 orchestrates M2 polarization of tumor-associated macrophages (TAMs) through the following regulatory mechanisms: activation of oncogenic signaling cascades (MAPK, AKT, STAT3, NF-κB, YAP1) (Sci Rep 14, 22541 (2024), Cells. 2022 Jan 11;11(2):229, Cancer Lett. 2025 Feb 1:610:217354) and stabilization of the immunomodulatory cytokine MIF (macrophage migration inhibitory factor) (Cell Death Dis. 2021 Feb 4;12(2):155). Targeting Hsp90 may provide a new therapeutic strategy to reverse the tumor-promoting phenotype of macrophages and impede tumors. Therefore, molecular docking was performed with Schrodinger to examine the binding mode and affinity of 12 with Hsp90 N-terminal domain. We obtained the top-ranked conformation of 12 with a docking score of -5.741 kcal/mol. Through the analysis of their binding mode (Figure 5 in the revised manuscript), it was found that 12 could form several hydrogen bonds with multiple amino acid residues such as Asp40, Asp42, Lys44, Asn92, and Thr171 on the HSP90 protein, further suggesting derivative 12 might targets HSP90. These results suggested that derivative 12 may exert antitumor effects by remodeling macrophage polarization through HSP90 targeting. Further target verification would be carried out in the near future.
- While the in vivo efficacy in the Hepa1-6 model is promising, additional tumor models should be considered to assess the broad applicability and efficacy of compound 12.
Response: We thank the reviewer for pointing this out. In this work, we employed compound 12 as a chemical probe to evaluate the antitumor effects of PAB derivatives through macrophage targeting. Cellular and in vivo experiments demonstrated its anti-tumor pharmacodynamic characteristics: effectively reversing macrophage pro-tumor polarization and counteracting tumor-promoting macrophage-mediated suppression of CD8+T cell proliferation and function. In macrophage-rich Hepa1-6 subcutaneous tumor models, compound 12 significantly inhibited tumor growth by remodeling the immunosuppressive tumor microenvironment. The results show that the PAB derivatives have the potential for further structural optimization and development.
In vivo pharmacodynamic evaluation showed mice remained healthy with no significant weight loss at 50 mg/kg, indicating good systemic tolerance. This dose demonstrated potent tumor growth suppression while maintaining safety, suggesting a favorable therapeutic window for the compound.
Notably, while compound 12 serves as the current active lead, structural optimization of this series remains essential to enhance potency and druggability. Subsequent development of improved derivatives will enable comprehensive pharmacodynamic profiling across diverse tumor models, PK/PD evaluation, accompanied by rigorous safety assessment - critical steps for advancing candidates toward translational application.
- Information on the PK/PD and toxicity profiles of compound 12 is lacking. These studies are crucial for the further development of the compound as a therapeutic agent.
Response: We thank reviewer for the suggestion. We performed in vivo PK studies for compound 12 in mice, and included the data in the revised manuscript.
- Either use CDCl3 or Chloroform-d. CNMR was done at a 700 MHz NMR spectrometer? (as it is shown as 176 MHz).
Response: We thank the reviewer for pointing this out. We used CDCl₃ instead of Chloroform-d in the revised manuscript. The ¹³C NMR frequency of 176 MHz corresponds to a 700 MHz instrument (since ¹³C resonates at ~1/4 the ¹H frequency).
